# An Analysis of Maternal, Social and Household Factors Associated with Childhood Anemia

**DOI:** 10.3390/ijerph18063105

**Published:** 2021-03-17

**Authors:** Vidya Chandran, Russell S. Kirby

**Affiliations:** College of Public Health, University of South Florida, Tampa, FL 33612, USA; vidyac1@usf.edu

**Keywords:** childhood anemia, National Family Health Survey, Madhya Pradesh

## Abstract

Anemia is highly prevalent in all strata of populations in India, with established evidence of intergenerational anemia. The state of Madhya Pradesh was selected to study childhood anemia as the population is mostly rural, with many tribal districts, and has the highest infant mortality rate in India. This study aims to understand the maternal, social and household factors that affect anemia among children aged 6 months to 5 years by analyzing the the National Family Health Survey (NFHS) conducted in 2015–2016. Children aged 6–59 months with estimated hemoglobin levels were included in this study. Bivariate and multivariable analyses were conducted to understand associations between childhood anemia and various socioeconomic factors. Two models to understand the presence of anemia and the levels of anemia were computed. Higher likelihood of having severe childhood anemia was observed among children of younger mothers (15- to 19-year-old mothers (adjusted odds ratio (aOR) 2.08, 95% confidence interval (CI): 1.06, 4.06, less educated (uneducated mothers aOR 2.25, 95% CI 1.13, 4.48) and belonged to a scheduled tribe (aOR 1.88, 95% CI 1.07, 3.29). Strong associations between anemia in mothers and their children suggest intergenerational anemia which has long-term effects. Malnourished children (severe stunting aOR 3.19, 95% CI 2.36, 4.31) and children born with very low birth weight (aOR 4.28, 95% CI 2.67, 6.87) were more likely to have anemia. These findings strongly suggest more proactive interventions including prenatal healthcare for women and monitoring of the nutrition children at the community level to combat childhood anemia. Evaluations of existing programs should be conducted to understand the gaps in reducing anemia and malnutrition in children.

## 1. An Analysis of Maternal, Social and Household Factors Associated with Childhood Anemia

### Background

Anemia and malnutrition are very common in the Indian population, and close associations between anemia and malnutrition have been established [1,2,3]. According to the World Health Organization, anemia is a condition in which the number of red blood cells (and consequently their oxygen-carrying capacity) is insufficient to meet the body’s physiological needs and anemia in children of 6–59 months of age is defined as a hemoglobin level less than 11 g per deciliter [4]. The most common cause of anemia is iron deficiency, and other causes include nutritional deficiencies, inflammation, parasitic infections, and certain inherited hemoglobinopathies [5,6,7,8].

Infants born to anemic women are undernourished and often turn anemic after a few months of birth [9,10]. Anemia in pregnant women is observed globally and the determinants for anemia in pregnancy are similar across the globe that include poverty, young mothers and poor iron intake [11,12,13]. Without timely intervention, these children grow up anemic and often marry early, giving birth to malnourished babies with less iron reserves who turn anemic within the first few months after birth [8,14,15,16]. There is a need to intervene in this intergenerational anemia to ensure good health of the overall population [10,14,15]. The Indian government has recognized the need for this intervention and has devised various strategies to combat the anemia and malnutrition; most of these efforts are reactive, such as anemia programs, establishment of nutritional rehabilitation centers, and umbrella efforts to better maternal and child health under the National Health Mission [17,18]. Initiatives such as National Nutritional Anemia Programme (NNACP), Weekly Iron and Folic Acid Supplementation (WIFS), National Iron Plus Initiative (NIPI) and Anemia Mukta Bharat Programme (AMBP) were aimed at reducing the incidence of anemia in women and children [19,20,21]. NNACP launched as early as in 1970, while NIPI and AMBP are more recent programs designed specifically to tackle anemia among the maternal and child health populations. AMBP was an intensified version of the existing NIPI program [19,22,23]. The rate of reduction of anemia due to all these efforts are not up to the mark [24]. There is a need to evaluate the public health system’s preventive and treatment efforts in combating anemia in India.

Socioeconomic inequality and malnutrition have been established in literature to be associated with anemia [25,26]. Socioeconomic inequality also arises from various discriminative categorizations among the populations through religious and caste classifications. Malnutrition is also related to cultural practices with respect to diet [27,28]. Anemia can prove to be a dangerous condition when it affects young children. It affects the child’s development in various ways such as physical and mental health, cognitive development and multiple morbidities [26,29]. There are health conditions that worsen the anemia in children [30,31]. In developing countries, the most prevalent form of anemia is iron deficiency anemia which is aggravated by worm infections, malaria and other infectious diseases such as human immunodeficiency virus (HIV) and tuberculosis [8,9,32].

Close associations between anemia in mothers and children have been established in the literature [9]. The major reasons for anemia in women can be categorized as intrinsic causes and extrinsic causes. Some of the intrinsic causes also result as an interplay of the extrinsic factors for prolonged periods of time [33,34,35]. Some of the extrinsic factors are poverty resulting in food insecurity and substandard hygiene practices, low status of women in the society and attitudes towards health and wellbeing such as hygiene and preventive healthcare [33,34]. One of the most important extrinsic causes of anemia and undernourishment in women is due to the low status of women in Indian society. The lack of education and empowerment, decision-making powers over income, lack of food and health security along with violence and preference for male children act against the good health of women and overall improvement of the country’s health. All these factors are closely associated with women’s anemia and malnutrition [28,33,34]. The male preference in families puts the female children on the backseat when it comes to education. Most of the girls either never go to school or have dropped out of school before completing their basic education [28,36,37]. Poverty and the dietary habits of the rural populations have been associated with malnutrition among rural populations. Male preference also exacerbates the food security of women within households [36].

Anemia and malnourishment in women are especially important since they pass it on to the next generation if there is no timely intervention [9]. In addition, when the women are married off early in life as commonly seen in rural populations, they conceive as anemic women and reinforce the continuation of this intergenerational anemia [38,39]. Undernourished and anemic women are more likely to give birth to babies who have low iron reserves [8,14,15]. Although the infants are not observed to be anemic during their early infancy, anemia starts becoming prevalent by the time infants turn a few months old. Even when the babies are exclusively breastfed, the breastmilk is low in nutrition and iron content since the mothers are anemic [8,14,15]. Such children are also observed to have low immunity and are prone to infections [38,40,41].

Two of the most important intrinsic causes are underlying conditions (e.g., sickle cell anemia) and deficiencies (e.g., iron, folate and vitamin B12) [26,29]. The children growing up in compromised extrinsic socioeconomic situations along with anemia and malnutrition are highly prone to infections and succumb to them [42]. Deaths due to diarrheal infections are high in India, especially in rural areas. The government has made efforts to avert such deaths by reactive interventions such as providing zinc and oral rehydration solution (ORS) to every household [37,38]. Undernutrition is associated with one-third of child deaths [43,44]. When children are severely malnourished, the chances of dying are high whereas when the children are mild or moderately malnourished, their chances of survival are high but suffer irreversible developmental damages [43,45]. According to Vir (2011), one in three children who die of pneumonia, diarrhea or other illnesses would survive if they are not malnourished.

## 2. Methodology

Madhya Pradesh is a state in central India with over half of the population in the lowest wealth quintiles (poorest and poorer). National Family Health Survey (NFHS) IV estimates for Madhya Pradesh [46] suggested that 68.9% of the children 6–59 months were anemic which is an improvement from 74% seen on NFHS-3 that was conducted in 2005–2006. The improvement from NFHS-3 to NFHS-4 among pregnant women (57.9% to 54.6%) was also observed to be similar [46]. Although the government of India and the state government have been making efforts to tackle the condition, it does not seem to be enough to overcome the situation among the maternal and child population. This study aims to understand the factors that affect anemia among children aged 6 months to 5 years in Madhya Pradesh by analyzing the NFHS-4.

Data for this study were taken from the Demographic and Health Survey (DHS) for India, NFHS IV, conducted in 2015–2016 by the International Institute for Population Sciences, Mumbai. The dataset used for this study was the birth recode (BR) (IABR74SV.ZIP) which consisted of the full birth history of all women interviewed including its information on pregnancy, postnatal care and immunization and health for children born in the last 5 years. The units of analysis (case) in this file are children born of eligible women and the data for the mother of each of these children is also included. Further information on the DHS and NFHS can be found at the DHS website [47].

The analyses were conducted on the BR dataset for the state of Madhya Pradesh for all the children whose hemoglobin levels were successfully estimated. Hemoglobin testing was performed using capillary blood and was undertaken through a portable battery-operated device called a HemoCue Hb 201 (HemoCue AB, Ängelholm, Sweden) + analyzer [47,48]. The consent was taken from a parent or any guardian responsible for the child. As per the standards of NFHS, children were considered anemic when their hemoglobin level was below 11.0 g/dL; mildly anemic (10.0–10.9), moderately anemic (7.0–9.9), severely anemic (less than 7.0). Hemoglobin levels were adjusted for altitude in enumeration areas that are above 1000 m [48], and the adjusted values were used for the analyses in this study. There were 62,803 women interviewed in the state of Madhya Pradesh and the final sample for this study consisted of 20,102 children aged 6 months to 59 months. The protocol for the survey was approved by the International Institute for Population Sciences (IIPS) Institutional Review Board and the ICF Institutional Review Board. The protocol was also reviewed by the U.S. Centers for Disease Control and Prevention (CDC) [46]. This secondary data analysis of DHS of India was reviewed by the University of South Florida IRB and determined exempt as not involving human subjects research.

### 2.1. Dependent and Independent Variables

In the first model, the dependent variable was the prevalence of anemia in children aged 6 months to 59 months in the state of Madhya Pradesh. The dependent variable was dichotomized as “anemic “and “Non-anemic”. In the second model, level of anemia (mild, moderate and severe anemia) was the dependent variable in a generalized logit model.

The independent variables analyzed consist of sociodemographic variables, maternal variables and child variables. Characteristics of the child included the age of the child, gender, birth weight and presence of stunting as a measure of nutrition, consuming any iron supplements and drugs for intestinal parasites. Variables measured across mothers included age of mother, her education and her anemia status. Sociodemographic variables included a composite variable of religion and caste, type of place of residence, wealth index and the type of sanitation facility at home. All the independent variables analyzed in this study were measured as categorical variables.

### 2.2. Analyses

Univariate and bivariate weighted analyses were carried out on the demographic and household characteristics of children aged 6–59 months. Weighted binary logistic regression was conducted to assess the effect of the independent variables on the presence of anemia in children under the age of 5 years. Furthermore, weighted multinomial logistic regression was carried out to assess the effect of these independent variables on the level of anemia among the children. Missing observations were deleted while computing the models. SAS^TM^ 9.4 (SAS Institute Inc., Cary, NC, USA) was used for the statistical analyses, and ArcMap 10.8 was used to develop the maps.

## 3. Results

The final analyses were carried out on a sample of 20,102 children in the age group of 6 months to 59 months in Madhya Pradesh. Table 1 and Table 2 show the descriptive statistics of the presence of anemia and the level of anemia in this study sample of children under the age of 5 years. Almost 70% (*n* = 14,015) of the sample was anemic. Of all the children who had anemia, more than half (54.7%, *n* = 7701) were in the moderate anemic category, 42.4% (*n* = 5898) in the mild category and 2.9% in the severe anemic category. Less than half of the children (42.6%, *n* = 8436) were born with a normal weight in the study sample, 34.5% (*n* = 6889) were born with a low birth weight (LBW) and 3.8% (*n* = 752) were born with a very low birth weight (VLBW). With respect to consumption of iron supplements and drugs for intestinal parasites, most children did not consume these drugs that could help in combating anemia: only 26% (*n* = 5367) consumed iron supplements and 29.7% (*n* = 6051) consumed drugs for intestinal parasites. With respect to the nutritional status of the children, most of the children (70.7%, *n* = 14,106) were normal weight with no wasting observed, whereas slightly more than half the population was not stunted (53.3%, *n* = −10,651).

Across the maternal age categories, most of the children (74.9%, *n* = 15,070) in the sample belonged to categories of mothers aged 20–29 years, and the highest prevalence of anemia was observed in this category as well. With respect to the maternal education, mothers of 40.9% children had secondary education. More than half the children had mothers who had some level of anemia (57.6%, *n* = 11,807). Three quarters of the population (*n* = 15,184) lived in rural areas and over 70% of the population in the rural areas (*n* = 10,729) had some form of anemia. With respect to hygiene practices, access to hygienic water was better than availability of hygienic sanitation facilities. Weighted percentages show that improved drinking water sources were available for 83.1% (*n* = 15,915) of the population whereas 62.4% (*n* = 12,180) of the population still had no toilet facility and used open areas for defecation. Most of the study sample consisted of children from the poor wealth quintiles (poorest 34.7%, *n* = 7174 and poorer 23.6%, *n* = 4733), and most of the anemic children belonged to these two quintiles.

Figure 1 shows the spatial distribution of the anemic children across the state of Madhya Pradesh at the district level. It can be observed from the figure that every district at least one-half of the children under 5 years of age as anemic. The higher prevalence of anemia can be observed to align with the south western districts in the tribal belt where moderate and severe forms of anemia are more prevalent. Higher prevalence of moderate and severe anemia can also be observed to coincide with the high priority districts (HPD) of Madhya Pradesh [49]. Of the 52 districts in Madhya Pradesh, 17 districts were identified as HPD based on a composite health index to prioritize maternal and child health interventions.

### Regression Results

Two models were computed to understand the presence of anemia and the level of anemia in children. The results of the models are presented on Table 3. A total of 1080 missing observations were deleted while computing the models: the two variables which majorly contributed to missing observations were sanitation facility and drinking water source. Respondents who were not *de jure* residents of the area of survey were excluded for these questions. While comparing the presence of anemia by maternal age, it was observed that children of younger women were more likely to have some form of anemia. Children of mothers in the age group 20–24 years were observed to be more likely to have anemia as compared to the 25–29 year-old mothers (adjusted odds ratio [aOR]: 1.23, 95% confidence interval [CI]: 1.13, 1.34). Similar association in younger women was observed in the multinomial regression analyzing the levels of anemia as well. Children of mothers in the age group of 15–19 years are more likely to have severe anemia as compared to mothers aged 25–29 years (aOR 2.08, 95% CI: 1.06, 4.06). Children of mothers aged 20–24 years were more likely to have mild (aOR: 1.2, 95% CI: 1.08, 1.32) and moderate (aOR: 1.26, 95% CI: 1.14, 1.39) anemia than 25- to 29-year-old mothers.

Children of less educated mothers appear to be more likely to have anemia than mothers with higher education. Children of mothers with no education were 2 times (aOR: 2.25, 95% CI: 1.13,4.48) more likely to have severe anemia as compared to children of mothers with an education of higher that secondary school education. Likelihood of presence of anemia decreased with increased level of education in the binary logistic model, and a similar pattern was observed in the odds ratios of mild and moderate anemia in the multinomial regression model as well.

Mother’s anemia status was significantly associated with child’s anemia in the binary regression model and the stratified multinomial model. Any level of anemia in the mothers was significantly associated with some form of anemia in children under the age of 5 years. Children of severely anemic mothers appear to be 5 times (aOR: 4.98, 95% CI: 2.29, 10.79) more likely to have with severe anemia as compared to non-anemic mothers. Similarly, children of mothers with mild and moderate anemia were also significantly associated with the presence of mild and moderate anemia in children. This aligns with the literature suggesting intergenerational anemia in India [10,14,16].

Among the variables associating the household variables to the presence of anemia in children, most variables were not significant. There was no significant association between the place of residence and the level of anemia in children in the adjusted models. There was no significant relationship between the level of anemia and the hygiene practices the household has access to such as improved water source and improved sanitation facilities. While analyzing the religion and caste, in scheduled tribes it was observed that higher caste Hindu children were more likely to have some form of anemia. This observation was significant in the binary and multinomial models.

Children born with LBW were significantly associated with a higher likelihood of having some form of anemia as compared to children born with a normal birth weight. Children born with VLBW (less than 1500 g) were more likely to have moderate or severe anemia: they were 4 times (aOR: 4.28, 95% CI: 2.67, 6.87) more likely to have severe anemia as compared to children born with a normal weight. Similarly, children born LBW (1500–2500 g) were more likely to have moderate anemia (aOR: 1.13, 95% CI: 1.02, 1.24). Consumption of iron supplement sand drugs for intestinal parasites appear to not have any significant association between the presence of anemia in children. Nutritional status was significantly associated with presence of anemia in children. Severely stunted children were 3 times (aOR: 3.19, 95% CI: 2.36, 4.31) more likely to have severe anemia as compared to children with no stunting.

## 4. Discussion

Anemia is very prevalent in India and Madhya Pradesh is among the Indian states with the highest prevalence of childhood anemia. The principal findings of the study demonstrate that maternal socioeconomic and health status play an important role in the health of their children. Most of the results align with the existing literature with respect to the socioeconomic factors [25,26]. Higher likelihood of having anemia was observed among mothers who were young, less educated and belonged to a scheduled tribe. This result aligns with the study that explores the caste based social inequalities and childhood anemia in India [50]. Caste systems and inequalities have been discriminatory and have affected the health of mothers and children for decades [28]. Strong associations between anemia in mothers and their children suggest intergenerational anemia which has chronic harmful effects on the child’s physical and mental health and development [10,14,16]. Teenage mothers were highly likely to have severe anemia. Even though the legal age to marry in India is 18 years for girls, marriage and childbirth before attaining 18 years of age is a common practice [8,15]. Such practices also hamper the continuation of education that puts women and infants at risk for adverse health outcomes. To combat such systemic practices in the communities, the frontline workers who are in communication with the girls in the villages need to be trained in prenatal health and education. One of the major factors to combat anemia in children is by combating anemia in women [9,38,39]. Efforts need to be made to strengthen existing programs such as iron supplementation programs in schools and through Rashtriya Kishore Swasthya Karyakram (RKSK), the national program for improving adolescent health. The inverse relationship of education and likelihood of having anemia shows that focusing on ensuring girl’s education will have long-term impacts in improving the overall health of the children. Evaluation of existing programs needs to be undertaken to understand the gaps in attaining the desired outcomes through efforts such as supplemental nutrition through Anganwadi Centers and weekly iron folic acid supplementation at schools. The health authorities need to take more proactive steps to combat chronic anemia in women along with the reactive efforts. There are widespread reactive efforts to improve the hemoglobin levels in pregnant women in Madhya Pradesh such as administering ferrous ascorbate for better iron absorption and iron sucrose administration. As a long-term solution to combat anemia at a larger scale, the authorities should focus on interventions with a systems approach [51].

In contrast to common beliefs that anemia is prevalent in rural populations with compromised hygiene practices [37,38,42], we found no association between rural residence or hygiene practices. Similarly, considering the government initiatives to provide iron supplements and drugs to control intestinal parasites for better absorption of iron for children, iron supplements or intestinal parasite drugs did not have any significant association on anemia in children [17,18,19,20]. This finding suggests that there needs to be a focus more on public health initiatives to combat anemia along with administering these drugs, which are one of the most common interventions taken by the government. The findings of the study reveal a higher prevalence of anemia among children from the lower socioeconomic strata. Most of the results align with the literature aside the type of place of residence factors [25,37,38,42]. There was no significance observed if the child lived in a rural area compared to an urban settling.

Children born with LBW and those with stunted growth had a higher prevalence of anemia. Stunting can be easily observed in children and could be used as an easy indicator to administer interventions. The current interventions taken by the government include the establishment of Nutritional Rehabilitation Centers (NRC) and providing supplemental nutrition through Anganwadi Centers (AWC) [52,53]. NRCs were first established in Madhya Pradesh and cater to children who are severe acute malnourished (SAM). The suggested treatment for SAM at the NRCs includes a 14-day inpatient care to improve the nutrition of the child. This facility-based management of SAM is promoted by reimbursing the daily wages of the parents since they would miss work while the child is admitted to the NRC. The high prevalence of anemia among such children throws light of the severe chronic malnutrition (SCM) as well. The NRC model is a vertical facility-based model that helps SAM children gain nutrition whereas to combat SCM more efforts need to be made at the community level to ensure that the children improve their nutrition while still at home. Community efforts such as those aiming at healthy diets in families can ensure that anemia in malnourished children is treated in a comprehensive manner. With many categories of frontline workers such as ASHAs, ANMs and Anganwadi workers, access is gained to such children even in the remotest areas in India. These workers can monitor the intervention efforts at the grassroots level and help bridge the gaps in ensuring good nutrition of the children. Strengthening the community efforts by monitoring and evaluation of the programs is essential for the envisioned outcomes. By tracking the children through a real-time approach such as the Jatak initiative [54] in Kerala that identified the malnourished children through a geographic information system (GIS), it would be beneficial to combat malnutrition systematically. The Jatak–Janani Nutrition Surveillance System was an effort begun in Attappadi, a tribal block of Kerala, and health and nutrition were monitored using a GIS-backed management information system. A similar effort was made in Jhabua district of Madhya Pradesh as well [53]. Large-scale implementation of such tracking at the community level could help identify high concentrations of malnourished and anemic children for targeting interventions.

Inherited and acquired hemoglobinopathies are also prevalent in Madhya Pradesh. Recent research has examined the higher prevalence of sickle cell anemia in the tribal belt in Madhya Pradesh [55]. Over half of the districts of Madhya Pradesh fall in areas with higher sickle cell prevalence which coincides also with the tribal belt [56]. Thalassemia is also prevalent among the tribal populations which also result in anemia in children [5,6,7,8]. Both these genetic conditions can be screened early on in the child’s life to prevent the low hemoglobin levels in the child’s life. Newborn screening and efficient follow up during the child’s life need to be undertaken more efficiently by the government in Madhya Pradesh. This effort can ensure that the children with these conditions are monitored regularly to reduce the morbidity as the child grows up.

Among study strengths, the DHS is a population-based survey and the weighted analyses ensure that the results are representative of the whole state. The survey that was analyzed was conducted in 2015–2016 (NFHS IV) and is the most recent NFHS data available as of now. NFHS V (2019–2020) is now in progress, with the data not yet available for analysis. The limitations of this study include the inability to understand the effect of child’s diet and malnutrition on anemia since the survey is limited with respect to the information available on the child’s diet. The authors also acknowledge the limitations of the study arising due to biases such as recall bias and reporting bias. Since the respondent who is the mother of the child is asked about events prior to the survey, there is a chance of recall bias. Also, since the survey is self-reported, there is a reporting bias. The study sample consisted of multiple children of the same mother, but we were unable to adjust for this clustering in the analysis, resulting in less precise estimates of standard errors and confidence intervals. To understand childhood anemia better, longitudinal studies could be undertaken to understand intergenerational anemia in the future. Such studies also could throw some light on the progress of anemia in relation to diet and the long-term effects of treatment.

## 5. Conclusions

In conclusion, this study explores the current scenario of childhood anemia in Madhya Pradesh, which needs a strategic and comprehensive approach to combat the highly prevalent condition. The strategies should ideally include a proactive approach with efficient surveillance of malnutrition and timely care. Since anemia has been a chronic condition usually present along with malnutrition, interventions focusing on healthy diets of the population is important. Training frontline workers to monitor these efforts at the grassroots level can help in bridging gaps in the interventions. Efforts to combat anemia in women could also help in improving anemia in children since the condition is seen as affecting different generations. Educating and empowering women could also help in improving the overall health of mothers and children.

In addition to these interventions, the government should improve efforts in the treatment and follow up of anemic children in the communities. Innovative strategies such as the Jatak–Janani Nutrition Surveillance System could assist in effective follow up of anemic children until the recovery and maintenance of good health. Lastly, systematic and periodic evaluation of the programs needs to be carried out to ensure effective reach of the interventions to every child in the state. These strategies combined with strengthening of the existing programs could also help in combating the morbidity and mortality associated with anemia and its complications in the state of Madhya Pradesh.

## Figures and Tables

**Figure 1 ijerph-18-03105-f001:**
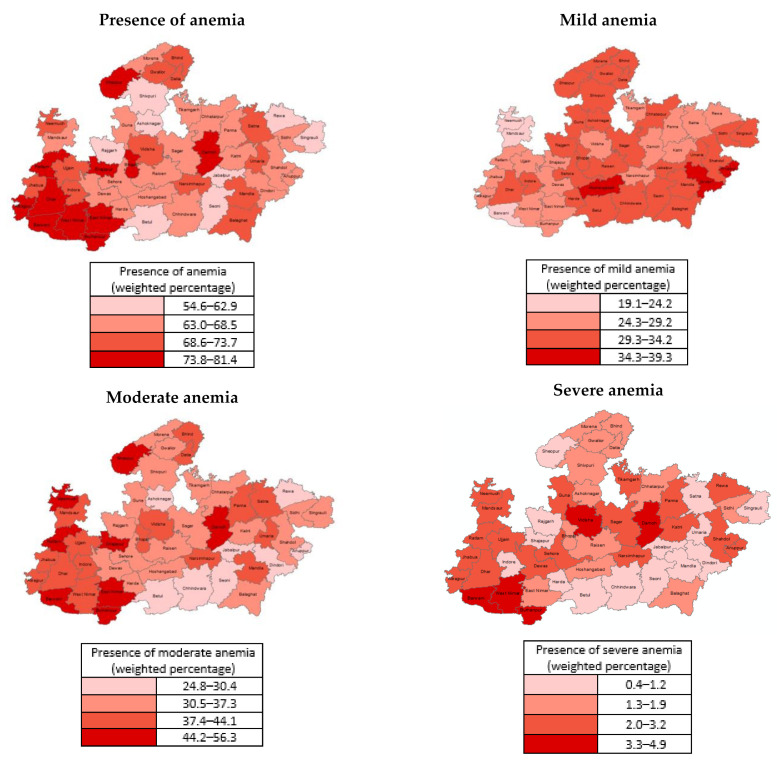
Spatial distribution of anemic children in Madhya Pradesh at the district-level.

**Table 1 ijerph-18-03105-t001:** Demographics and household characteristics of children under aged 6–59 months in Madhya Pradesh by anemia status.

Characteristic	Total (*n* = 20,102)		Anemic (*n* = 14,015)		Non Anemic (*n* = 6087)	
	*n*	Weighted %	*n*	Weighted %	*n*	Weighted %
Current age of the mother						
15–19 years	368	1.8	293	2.0	75	1.2
20–24 years	7116	35.6	5304	38.3	1812	29.4
25–29 years	7954	39.3	5302	37.5	2652	43.3
30–34 years	3272	16.4	2214	15.7	1058	17.9
35–39 years	1035	5.2	662	4.8	373	6.1
40–44 years	267	1.3	189	1.3	78	1.3
45–49 years	90	0.4	51	0.3	39	0.7
Mother’s education						
No education	7276	34.3	5271	35.6	2005	31.4
Primary	3794	18.6	2687	18.8	1107	17.9
Secondary	7840	40.9	5340	40.2	2500	42.4
Higher	1192	6.3	717	5.4	475	8.3
Mother’s anemia status						
Not anemic	8225	42.4	5202	38.7	3023	50.8
Mild	8395	41.6	6079	43.2	2316	37.9
Moderate	3181	14.9	2502	16.9	679	10.5
Severe	231	1.1	179	1.2	52	0.8
Missing: 0.35% (*n* = 70)						
Place of residence						
Rural	15,187	74.1	10,729	75.2	4458	71.7
Urban	4915	25.9	3286	24.8	1629	28.3
Sanitation facility ^$^						
Improved sanitation facility	6630	35.9	4398	34.3	2232	39.4
Not improved sanitation facility	331	1.8	231	1.7	100	1.8
No toilet facility-Bush/fields	12,180	62.4	8712	64.0	3468	58.9
Missing: 5.02% (*n* = 961)						
Drinking water source ^#^						
Improved water source	15,915	83.1	11,056	82.9	4859	83.6
No improved water source	3226	16.9	2285	17.1	941	16.4
Missing: 5.02% (*n* = 961)						
Religion and caste						
Hindu high caste	1876	10.0	1220	9.4	656	11.3
Hindu lower caste	10,928	57.2	7372	55.6	3556	60.7
Hindu ST ^a^	5203	23.2	3973	25.5	1230	18.0
Hindu others	351	1.7	246	1.8	105	1.5
Muslim	1597	7.1	1111	6.9	486	7.5
Others	147	0.8	93	0.7	54	0.9
Wealth quintile						
Poorest	7174	34.7	5217	36.3	1957	31.1
Poorer	4733	23.6	3351	23.8	1382	22.9
Middle	3133	15.7	2188	15.6	945	15.9
Richer	2706	13.7	1780	13.0	926	15.2
Richest	2356	12.4	1479	11.3	877	15.0
Gender of the child						
Female	9592	47.9	6663	47.7	2929	48.2
Male	10,510	52.1	7352	52.3	3158	51.8
Age of the child						
Less than 1 year old	2222	11.1	1667	12.0	555	8.9
1–2 years old	4413	21.9	3598	25.8	815	13.1
2–3 years old	4469	22.3	3326	23.7	1143	19.1
3–4 years old	4550	22.7	2920	20.9	1630	26.6
4–5 years old	4448	22.1	2504	17.6	1944	32.2
Birthweight of the child						
VLBW *	752	3.8	576	4.3	176	2.8
LBW *	6889	34.5	4857	34.9	2032	33.5
Normal weight *	8436	42.6	5713	41.3	2723	45.6
Overweight *	234	1.2	143	1.1	91	1.6
Not weighed at birth	3145	14.5	2266	15.0	879	13.2
Don’t know	646	3.4	460	3.4	186	3.3
Taking iron pills, sprinkles or syrup						
Yes	5367	26.0	3749	25.9	1618	26.3
No	14,579	73.2	10,158	73.3	4421	73.0
Don’t know	156	0.8	108	0.8	48	0.7
Drugs for intestinal parasites in last 6 months						
Yes	6051	29.7	4212	29.5	1839	30.1
No	13,889	69.4	9693	69.6	4196	69.0
Don’t know	162	0.9	110	0.9	52	0.9
Presence of stunting						
Not stunted	10,651	53.3	7020	50.0	3631	60.5
Flagged for height for age	916	4.4	688	4.8	228	3.3
Moderately or severely stunted	4711	23.6	3322	23.9	1389	22.7
Severely stunted	3824	18.8	2985	21.2	839	13.5
Presence of wasting						
Normal weight/no wasting	14,106	70.7	9697	69.4	4409	73.6
Flagged	916	4.4	688	4.8	228	3.3
Moderately	3151	15.5	2229	15.8	922	14.8
Severely wasted	1676	8.1	1214	8.5	462	7.3
Overweight	253	1.4	187	1.5	66	1.1

All the % are weighted percentages in this table. * VLBW: Less than 1500 g, LBW: 1500–2500 g, Normal weight: 2500–4000 g, Overweight: 4000+ grams. ^$^ Improved sanitation facilities are those where there is flush to piped sewer system, flush to septic tank, flush to pit latrine, ventilated improved pit latrine, pit latrine with slab and composting toilet. ^#^ Improved drinking water sources are piped into dwelling, piped to yard/plot, public tap/standpipe, tube well or borehole, protected well, protected spring, rainwater and community reverse osmosis plant. not improved drinking water sources include unprotected well, unprotected spring, river/dam/lake/ponds/stream/canal/irrigation channel, tanker truck, cart with small tank and bottled water. ^a^ ST: Scheduled tribe.

**Table 2 ijerph-18-03105-t002:** Demographics and household characteristics of children under aged 6–59 months in Madhya Pradesh by anemia severity.

Characteristic	Mild Anemic (*n* = 5898) Hb 10–10.9 g/dL		Moderately Anemic (*n* = 7701) Hb 7.1–9.9 g/dL		Severely Anemic (*n* = 416) Hb < 7.1 g/dL	
	*n*	%	*n*	%	*n*	%
Current age of the mother						
15–19 years	103	1.7	172	2.2	18	4.4
20–24 years	2087	35.8	3056	40.1	161	40.2
25–29 years	2300	38.6	2859	36.9	143	33.6
30–34 years	997	16.8	1147	14.9	70	15.8
35–39 years	306	5.4	338	4.3	18	4.6
40–44 years	83	1.4	101	1.3	5	1.3
45–49 years	22	0.3	28	0.3	1	0.1
Mother’s education						
No education	2084	33.5	3004	36.8	183	43.9
Primary	1152	19.2	1466	18.7	69	16.4
Secondary	2306	41.2	2887	39.6	147	35.6
Higher	356	6.1	344	4.9	17	4.2
Mother’s anemia status						
Not anemic	49	0.9	119	1.4	11	2.1
Mild	863	14.0	1515	18.6	124	27.6
Moderate	2555	43.1	3362	43.5	162	40.0
Severe	2405	42.0	2680	36.5	117	30.2
Missing: 0.35% (*n* = 70)						
Place of residence						
Rural	4517	75.3	5909	75.2	303	71.9
Urban	1381	24.7	1792	24.8	113	28.1
Sanitation facility						
Improved sanitation facility	1943	35.9	2320	33.1	135	33.4
Not improved sanitation facility	88	1.5	136	1.9	7	1.8
No toilet facility-Bush/fields	3589	62.6	4867	65.0	256	64.9
Missing: 5.02% (*n* = 961)						
Drinking water source						
Improved drinking water source	4633	82.6	6091	83.1	332	82.9
No improved drinking water source	987	17.4	1232	16.9	66	17.1
Missing: 5.02% (*n* = 961)						
Religion and caste						
Hindu high caste	542	10.1	651	9.0	27	7.2
Hindu lower caste	3205	56.3	3958	55.1	209	54.1
Hindu ST	1588	24.5	2256	26.2	129	28.8
Hindu others	108	1.8	132	1.7	6	1.6
Muslim	409	6.4	659	7.3	43	8.0
Others	46	0.8	45	0.7	2	0.3
Wealth quintile						
Poorest	2125	35.2	2941	37.2	151	35.5
Poorer	1385	23.3	1861	24.2	105	25.5
Middle	907	15.3	1205	15.6	76	19.2
Richer	765	13.5	970	12.8	45	9.6
Richest	716	12.6	724	10.3	39	10.3
Gender of the child						
Female	3001	50.4	4094	53.2	257	60.9
Male	2897	49.6	3607	46.8	159	39.1
Age of the child	5898					
Less than 1 year old	1176	19.9	2262	29.8	160	37.9
1–2 years old	1299	22.4	1919	24.6	108	25.2
2–3 years old	1405	23.8	1460	19.1	55	14.0
3–4 years old	1381	23.1	1083	13.7	40	9.4
4–5 years old	637	10.8	977	12.9	53	13.5
Birthweight of the child						
VLBW *	197	3.6	338	4.5	41	10.4
LBW *	2010	34.1	2703	35.7	144	33.4
Normal weight *	2544	43.6	3036	39.9	133	32.7
Overweight *	54	1.0	82	1.1	7	2.0
Not weighed at birth	903	14.5	1289	15.3	74	17.8
Don’t know	190	3.3	253	3.5	17	3.6
Taking iron pills, sprinkles or syrup						
Yes	1580	25.9	2056	25.9	113	25.5
No	4278	73.4	5582	73.3	298	73.2
Don’t know	40	0.7	63	0.8	5	1.3
Drugs for intestinal parasites in last 6 months						
Yes	1777	29.6	2296	29.3	139	32.6
No	4065	69.2	5353	70.0	275	66.9
Don’t know	56	1.1	52	0.7	2	0.5
Presence of stunting						
Not stunted	3227	54.7	3648	47.1	145	36.8
Flagged for height for age	268	4.5	385	4.9	35	8.6
Moderately or severely stunted	987	16.6	1858	24.3	140	30.9
Severely stunted	1416	24.2	1810	23.7	96	23.7
Presence of wasting						
Normal weight/no wasting	4087	69.8	5334	69.4	276	65.5
Flagged	268	4.5	385	4.9	35	8.6
Moderately	933	15.8	1237	15.8	59	13.8
Severely wasted	534	8.6	637	8.3	43	11.2
Overweight	76	1.3	108	1.6	3	1.0

All the % are weighted percentages in this table. * VLBW: Less than 1500 g, LBW: 1500–2500 g, Normal weight: 2500–4000 g, Overweight: 4000+ grams.

**Table 3 ijerph-18-03105-t003:** Adjusted odds ratios (aORs) of the weighted analyses representing association of anemia in children and sociodemographic characteristics (Model 1: Presence of anemia, Model 2: Level of anemia).

Sociodemographic Characteristics	Model 1 (Anemia vs. No Anemia)	Model 2 (Levels of Anemia vs. No Anemia)		
	Anemic (Hb < 11 g/dL) aOR (95% CI)	Mild Anemic (Hb 10–10.9 g/dL) aOR (95% CI)	Moderately Anemic (Hb 7.1–9.9 g/dL) aOR (95% CI)	Severely Anemic (Hb < 7.1 g/dL) aOR (95% CI)
Current age of the mother				
15–19 years	1.19 (0.88, 1.60)	1.19 (0.84, 1.70)	1.14 (0.84, 1.56)	**2.08 (1.06, 4.06)**
20–24 years	**1.23 (1.13, 1.34)**	**1.20 (1.08, 1.32)**	**1.26 (1.14, 1.39)**	1.28 (0.97, 1.69)
25–29 years	1.00 (ref)	1.00 (ref)	1.00 (ref)	1.00 (ref)
30–34 years	1.10 (0.99, 1.22)	1.10 (0.98, 1.24)	1.09 (0.97, 1.23)	1.31 (0.90, 1.89)
35–39 years	0.98 (0.83, 1.16)	1.04 (0.86, 1.25)	0.92 (0.75, 1.13)	1.05 (0.58, 1.90)
40–44 years	1.22 (0.89, 1.67)	1.19 (0.85, 1.65)	1.26 (0.86, 1.85)	1.31 (0.47, 3.65)
45–49 years	0.71 (0.46, 1.10)	0.66 (0.38, 1.16)	0.78 (0.46, 1.32)	0.31 (0.04, 2.55)
Mother’s education				
No education	**1.54 (1.27, 1.87)**	**1.36 (1.10, 1.69)**	**1.69 (1.33, 2.15)**	**2.25 (1.13, 4.48)**
Primary	**1.53 (1.26, 1.86)**	**1.45 (1.17, 1.80)**	**1.61 (1.27, 2.05)**	1.58 (0.78, 3.22)
Secondary	**1.32 (1.11, 1.58)**	**1.27 (1.04, 1.55)**	**1.38 (1.11, 1.72)**	1.27 (0.66, 2.45)
Higher	1.00 (ref)	1.00 (ref)	1.00 (ref)	1.00 (ref)
Mother’s anemia status				
Not anemic	1.00 (ref)	1.00 (ref)	1.00 (ref)	1.00 (ref)
Mild	**1.43 (1.33, 1.55)**	**1.33 (1.21, 1.45)**	**1.53 (1.40, 1.67)**	**1.65 (1.22, 2.23)**
Moderate	**1.98 (1.76, 2.22)**	**1.56 (1.37, 1.78)**	**2.32 (2.04, 2.64)**	**4.18 (3.00, 5.82)**
Severe	**1.85 (1.28, 2.67)**	1.25 (0.82, 1.92)	**2.37 (1.58, 3.56)**	**4.98 (2.29, 10.79)**
Place of residence				
Rural	0.98 (0.87, 1.11)	1.06 (0.93, 1.21)	0.93 (0.81, 1.07)	0.73 (0.46, 1.16)
Urban	1.00 (ref)	1.00 (ref)	1.00 (ref)	1.00 (ref)
Sanitation facility				
Improved sanitation facility	1.00 (ref)	1.00 (ref)	1.00 (ref)	1.00 (ref)
Not improved sanitation facility	0.92 (0.69, 1.22)	0.82 (0.61, 1.11)	1.01 (0.72, 1.41)	0.87 (0.34, 2.21)
No toilet facility-Bush/fields	0.92 (0.81, 1.04)	0.91 (0.79, 1.06)	0.93 (0.81, 1.07)	0.84 (0.56, 1.27)
Drinking water source				
Improved drinking water source	1.06 (0.96, 1.18)	1.03 (0.92, 1.15)	1.09 (0.96, 1.23)	1.05 (0.76, 1.46)
No improved drinking water source	1.00 (ref)	1.00 (ref)	1.00 (ref)	1.00 (ref)
Religion and caste				
Hindu high caste	1.00 (ref)	1.00 (ref)	1.00 (ref)	1.00 (ref)
Hindu lower caste	0.92 (0.80, 1.05)	0.93 (0.80, 1.08)	0.90 (0.76, 1.06)	1.19 (0.73, 1.92)
Hindu ST	**1.31 (1.11, 1.54)**	**1.31 (1.10, 1.56)**	**1.29 (1.06, 1.56)**	**1.88 (1.07, 3.29)**
Hindu others	1.22 (0.89, 1.68)	1.26 (0.87, 1.82)	1.18 (0.83, 1.68)	1.38 (0.50, 3.83)
Muslim	0.94 (0.77, 1.14)	0.86 (0.69, 1.07)	1.00 (0.79, 1.26)	1.27 (0.66, 2.44)
Others	0.96 (0.62, 1.49)	1.04 (0.65, 1.67)	0.89 (0.51, 1.54)	0.71 (0.15, 3.37)
Wealth quintile				
Middle	1.07 (0.90, 1.28)	0.97 (0.80, 1.19)	1.17 (0.94, 1.44)	1.29 (0.71, 2.34)
Poorer	1.08 (0.89, 1.31)	1.00 (0.80, 1.24)	1.17 (0.93, 1.47)	1.05 (0.55, 2.00)
Poorest	1.1 (0.90, 1.35)	1.01 (0.80, 1.27)	1.22 (0.95, 1.55)	0.86 (0.43, 1.72)
Richer	0.97 (0.83, 1.13)	0.93 (0.78, 1.11)	1.02 (0.84, 1.24)	0.68 (0.39, 1.18)
Richest	1.00 (ref)	1.00 (ref)	1.00 (ref)	1.00 (ref)
Gender of the child				
Female	1 (0.93, 1.08)	1.07 (0.99, 1.16)	0.96 (0.89, 1.04)	**0.71 (0.56, 0.90)**
Male	1.00 (ref)	1.00 (ref)	1.00 (ref)	1.00 (ref)
Age of the child				
Less than 1 year old	**2.61 (2.27, 3.00)**	**1.75 (1.48, 2.06)**	**3.73 (3.17, 4.38)**	**5.86 (3.63, 9.48)**
1–2 years old	**3.59 (3.19, 4.05)**	**2.13 (1.85, 2.45)**	**5.45 (4.76, 6.24)**	**9.71 (6.39, 14.74)**
2–3 years old	**2.25 (2.02, 2.49)**	**1.63 (1.44, 1.84)**	**3.01 (2.66, 3.42)**	**4.50 (2.96, 6.85)**
3–4 years old	**1.44 (1.30, 1.60)**	**1.26 (1.12, 1.43)**	**1.67 (1.46, 1.90)**	**1.73 (1.12, 2.70)**
4–5 years old	1.00 (ref)	1.00 (ref)	1.00 (ref)	1.00 (ref)
Birthweight of the child				
VLBW (Less than 1500 g)	**1.48 (1.20, 1.83)**	1.22 (0.95, 1.56)	**1.64 (1.30, 2.06)**	**4.28 (2.67, 6.87)**
LBW (1500–2500 g)	1.07 (0.99, 1.17)	1.01 (0.92, 1.11)	**1.13 (1.02, 1.24)**	1.31 (1.00, 1.73)
Normal weight (2500–4000 g)	1.00 (ref)	1.00 (ref)	1.00 (ref)	1.00 (ref)
Overweight (4000+ grams)	0.8 (0.58, 1.11)	0.70 (0.46, 1.06)	0.86 (0.59, 1.23)	2.37 (0.94, 6.00)
Not weighed at birth	1.06 (0.94, 1.19)	1.00 (0.88, 1.14)	1.10 (0.96, 1.27)	1.51 (1.03, 2.22)
Don’t know	1.13 (0.92, 1.40)	1.01 (0.79, 1.30)	1.24 (0.97, 1.59)	1.67 (0.92, 3.04)
Taking iron pills, sprinkles or syrup				
Don’t know	1.06 (0.69, 1.61)	0.91 (0.56, 1.48)	1.17 (0.70, 1.96)	2.55 (0.93, 7.02)
Yes	1.00 (ref)	1.00 (ref)	1.00 (ref)	1.00 (ref)
No	1.04 (0.94, 1.15)	1.06 (0.94, 1.18)	1.02 (0.91, 1.15)	1.10 (0.81, 1.51)
Drugs for intestinal parasites in last 6 months				
Don’t know	1.04 (0.70, 1.53)	1.44 (0.93, 2.22)	0.75 (0.45, 1.24)	0.33 (0.08, 1.32)
Yes	1.00 (ref)	1.00 (ref)	1.00 (ref)	1.00 (ref)
No	1.01 (0.92, 1.11)	1.00 (0.90, 1.12)	1.02 (0.92, 1.13)	0.86 (0.65, 1.15)
Presence of stunting				
Not stunted	1.00 (ref)	1.00 (ref)	1.00 (ref)	1.00 (ref)
Flagged for height for age	1.66 (0.87, 3.16)	1.28 (0.62, 2.64)	**2.19 (1.03, 4.66)**	0.67 (0.10, 4.61)
Moderately or severely stunted	**1.27 (1.16, 1.39)**	**1.17 (1.05, 1.29)**	**1.36 (1.23, 1.51)**	**1.66 (1.22, 2.27)**
Severely stunted	**1.73 (1.55, 1.94)**	**1.28 (1.12, 1.46)**	**2.14 (1.89, 2.43)**	**3.19 (2.36, 4.31)**
Presence of wasting				
Normal weight/no wasting	1.00 (ref)	1.00 (ref)	1.00 (ref)	1.00 (ref)
Flagged	0.92 (0.49, 1.75)	1.13 (0.55, 2.31)	0.71 (0.34, 1.50)	4.86 (0.71, 33.48)
Moderately or severely wasted	1.03 (0.93, 1.14)	1.07 (0.95, 1.21)	1.00 (0.89, 1.12)	0.90 (0.63, 1.28)
Overweight	1.36 (0.93, 2.00)	1.37 (0.92, 2.04)	1.39 (0.86, 2.25)	0.85 (0.25, 2.87)

All bolded values are significant at *p*-value < 0.05.

## Data Availability

The data are available upon request for download through https://dhsprogram.com (accessed on 28 December 2020).

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
