# Peer review of "An Analysis of Maternal, Social and Household Factors Associated with Childhood Anemia"

_ijerph, 2021, doi:10.3390/ijerph18063105_

Round 1

Reviewer 1 Report

Overall, this is a quality article highlighting important variables that contribute to anemia in young children. I only have a few comments:

For the entire article, there is a lack of references. Every statement made referring to other work should have a reference. 

Line 30-31: grammar and provides references

Line 105: Define ORS

Methodology

Provide a reference for the hemoglobin/altitude adjustment. Also, address the method you chose; this needs to be transparent. It also might be worth considering stratifying by altitude level as current data suggests higher altitudes have considerably more impact on Hb than lower altitudes.

Define LBW and VLBW. The stick to LBW and VLBW throughout manuscript after you define; i.e. low birth weight (LBW).

Results

In your tables, please indicate what all symbols reference and mean and why some items are bolded. This should all be in the table legend.

Line 236-237: provide references when you make statements like these.

Author Response

Please see the attachment- it contains responses to all reviewer comments

Reviewer 2 Report

The manuscript entitled “An analysis of maternal, social and household factors associated with childhood anemia” presents interesting issue, but some areas must be corrected.

Major:

Authors planned an interesting issue to be studied, namely factors determining anemia and malnutrition development, but unfortunately, they did not include to their analysis the most important determinant of both anemia and malnutrition – which is diet that is followed. As indicated by WHO (https://www.who.int/nutrition/publications/en/ida_assessment_prevention_control.pdf) the dietary iron deficiency is here the key determinant (not only the iron supplementation, but exactly iron intake), so if it was not studied, it is hard even to try to conclude. Not including this factor to analysis is a major source of bias which causes that Authors may have obtained the false results. Taking this into account, we can not perceive their results as valid.

General:

The manuscript should be formatted according to the instructions for authors (e.g. sections).

Abstract:

Madhya Pradesh should be defined as a state

Authors should present specific numeric results of the study accompanied by the numeric results of their statistical analysis (e.g. p-Value)

Introduction:

Authors should prepare this section not only to be interesting for readers from India, but to be interesting for international readers. If Authors prepare their manuscript only for their national readers, they should publish it in some national journal. So, Authors should broaden the information presenting here more international data from various countries, not only the ones from India.

Authors should focus on the general factors contributing anemia development and blood iron level which were studied in other research (mainly those conducted in a large cohorts of women) – Authors should get familiar with e.g. https://pubmed.ncbi.nlm.nih.gov/15819466/; https://pubmed.ncbi.nlm.nih.gov/31083370/; https://pubmed.ncbi.nlm.nih.gov/32680488/.

Lines 112-120 – the justification of the studied population should be rather presented in materials and Methods Section

Materials and Methods:

For the presented study it is necessary to obtain the ethical committee agreement. Not only the study was conducted in a group of human subjects, and their personal data are obtained, but also some biochemical analysis (haemoglobin level) must be approved by the committee. In the presented information about the project, there was no information about ethical committee agreement, so it is not specified if Author did not obtain such agreement, or obtained, but did not provide information about it.

Authors should precisely describe applied tools, including questionnaires, while indicating if they were validated, in what population were they validated and what were the results of validation. Authors included some questions about issues which may be perceived as sensitive ones (e.g. religion, sanitation facility, wealth, intestinal parasites, etc.), so depending on the applied tool and applied methodology of data gathering, they may have obtained the false results.

There are the major methodological problems (see above)

Results:

Authors should verify the representativeness of the studied group

Authors should not reproduce in the text data that are already presented in tables

Tables should be stand-alone ones – be able to be understand without reading the manuscript, so Authors should explain everything needed in footnotes.

Figure 1 – does not present any essential data for international readers

Discussion:

Authors should in their discussion include 3 areas: (1) compare gathered data with the results by other authors, (2) formulate implications of the results of their study and studies by other authors, (3) formulate the future areas which should be studied

Authors should discuss all the limitations of the study (see above)

Authors should formulate real conclusions from their study indicating what can be concluded, not only the summery of the study.

Authors Contributions:

Should be presented

Author Response

Please see the attachment, which has responses to all reviewer comments.

Round 2

Reviewer 2 Report

The manuscript entitled “An analysis of maternal, social and household factors associated with childhood anemia” presents interesting issue, but some areas must be corrected.

Major:

Authors planned an interesting issue to be studied, namely factors determining anemia and malnutrition development, but unfortunately, they did not include to their analysis the most important determinant of both anemia and malnutrition – which is diet that is followed. As indicated by WHO (https://www.who.int/nutrition/publications/en/ida_assessment_prevention_control.pdf) the dietary iron deficiency is here the key determinant (not only the iron supplementation, but exactly iron intake), so if it was not studied, it is hard even to try to conclude. Not including this factor to analysis is a major source of bias which causes that Authors may have obtained the false results. Taking this into account, we can not perceive their results as valid. This problem should be reflected i whole manuscript (all the sections – including Abstract, Materials and Methods, Discussion and Conclusion), as without it readers may have false image of the situation.

General:

The manuscript should be formatted according to the instructions for authors

Materials and Methods:

Authors should precisely describe applied tools, including questionnaires, while indicating if they were validated, in what population were they validated and what were the results of validation. Authors included some questions about issues which may be perceived as sensitive ones (e.g. religion, sanitation facility, wealth, intestinal parasites, etc.), so depending on the applied tool and applied methodology of data gathering, they may have obtained the false results.

There are the major methodological problems (see above)

Results:

Authors should verify the representativeness of the studied group

Authors should not reproduce in the text data that are already presented in tables

Tables should be stand-alone ones – be able to be understand without reading the manuscript, so Authors should explain everything needed in footnotes.

Figure 1 – does not present any essential data for international readers

Discussion:

Authors should discuss all the limitations of the study (see above)

Authors should formulate real conclusions from their study indicating what can be concluded, not only the summery of the study.

Authors Contributions:

Should be presented

Author Response

We have updated our comments in response to the reviewer comments.  We feel that we have addressed all the issues, including formatting the paper and references as requested.  We would like to resubmit our revised manuscript.
